# Guinea Pig X Virus Is a Gammaherpesvirus

**DOI:** 10.3390/v17081084

**Published:** 2025-08-05

**Authors:** Vy Ngoc Yen Truong, Robert Ellis, Brent A. Stanfield

**Affiliations:** 1Department of Veterinary Biomedical Sciences, School of Veterinary Medicine, Seoul National University, Gwanak-gu, Seoul 08826, Republic of Korea; 2Department of Pathobiological Sciences, School of Veterinary Medicine, Louisiana State University, Baton Rouge, LA 70803, USA

**Keywords:** Guinea Pig X Virus, gammaherpesvirus, viral evolution, host adaptation, herpesviridae, viral pathogenesis, sequence similarity, herpesviruses

## Abstract

The Guinea Pig X Virus (GPXV), a newly identified gammaherpesvirus, provides an opportunity to study viral evolution and host–virus dynamics. This study characterizes the GPXV genome and investigates its phylogenetic relationships and divergence from related viruses through comparative genomic and phylogenetic analyses. Virus propagation was conducted in Vero cells, followed by genomic DNA extraction and pan-herpesvirus nested PCR. Sanger sequencing filled gaps in the initial genome assembly, and whole-genome sequencing was performed using the Illumina MiSeq platform. Phylogenetic analyses focused on ORF8 (glycoprotein B), ORF9 (DNA polymerase catalytic subunit), ORF50 (RTA: replication and transcription activator), and ORF73 (LANA: latency-associated nuclear antigen). Results showed that GPXV ORFs showed variable evolutionary relationships with other gammaherpesviruses, including divergence from primate-associated viruses and clustering with bovine and rodent viruses. In addition to phylogenetics, a comprehensive comparative analysis of protein-coding genes between GPXV and the previously described Guinea Pig Herpes-Like Virus (GPHLV) revealed divergence. Twenty-four non-ORF genomic features were unique to GPXV, while 62 shared ORFs exhibited low to high sequence divergence. These findings highlight GPXV’s distinct evolutionary trajectory and its potential role as a model for studying host-specific adaptations and gammaherpesvirus diversity.

## 1. Introduction

Guinea pigs have long been valuable animal models for studying human diseases, particularly in virology [1]. They are widely used in research on respiratory viruses, tuberculosis, and herpesvirus infections due to their similarities to human immune responses and the progression of infections in the body [1,2,3,4,5,6]. In herpesvirus studies, guinea pigs serve as an effective model for investigating viral pathogenesis, latency, and vaccine development, especially in the context of congenital cytomegalovirus (CMV) transmission and antiviral testing [7,8,9,10,11].

A study has demonstrated the effectiveness of antiviral compounds in treating cutaneous herpesvirus infections in guinea pigs [1]. Furthermore, the isolation of Guinea Pig X Virus (GPXV), a novel herpesvirus endemic to guinea pigs, has been characterized as distinct from other known guinea pig herpesviruses, such as Guinea Pig Cytomegalovirus (GPCMV) and Guinea Pig Herpes-like Virus (GPHLV) [12].

There are three known guinea pig herpesviruses: Guinea Pig Cytomegalovirus (GPCMV), Guinea Pig Herpes-like Virus (GPHLV), and Guinea Pig X Virus (GPXV). GPXV was initially isolated from buffy coat cells co-cultured with whole guinea pig embryo (GPE) cells in adult strain 2 guinea pigs. This virus appears to correspond to the guinea pig herpesvirus first described by Smith et al. in 1980 [1]. However, because no genomic sequence was available from that isolation, a direct genetic comparison is not possible. The virus presented in this study, which we refer to as GPXV, represents the first complete genome characterization of a gammaherpesvirus isolated from guinea pigs, consistent with the earlier description. While GPXV shares the general size, morphology, and biochemical properties typical of the herpesvirus group, it is serologically distinct from both GPCMV and GPHLV [1].

The further characterization of GPXV, including buoyant density measurements and restriction endonuclease cleavage patterns, emphasizes its distinct genetic makeup compared to other endogenous guinea pig herpesviruses [1]. These analyses reveal significant differences in base composition and nucleotide sequence [1]. Preliminary investigations into GPXV pathogenesis in Hartley guinea pigs mainly suggest a prolonged period of viremia, contrasting with the kinetics observed in experimental GPCMV infection [1]. While viremia persists for weeks following GPXV inoculation, natural and experimental GPHLV infection is characterized by the continuous, persistent virus infection of circulating leukocytes [1].

GPXV also shares significant genomic similarities with other members of the Gammaherpesvirinae subfamily, such as Bat gammaherpesvirus 2 (BtGHV2), Marmot herpesvirus 1 (MhHV1), and Saimiriine gammaherpesvirus 2 (SaHV2) [13]. These viruses belong to the Rhadinovirus genus within the Gammaherpesvirinae subfamily of the Herpesviridae family, a group known for infecting a wide range of mammals and establishing long-term latent infections. Like BtGHV2, GPXV encodes a viral IL-10 homolog, an immunomodulatory protein that aids in immune evasion and viral persistence [14]. It also shows genomic similarity to MhHV1, suggesting that GPXV could serve as a model virus—similar to how Woodchuck hepatitis virus (WHV) is used to study Hepatitis B virus pathogenesis [15]. Furthermore, GPXV shares similarities to SaHV2, including conserved gene synteny, sequence homology, and shared regulatory motifs [14,16]. Both viruses display lymphoid cell tropism, modulate host immune responses, and establish latent infections, showing conserved pathogenic and immunoevasive strategies among gammaherpesviruses [16].

Previously, we described the genome of GPHLV, a gammaherpesvirus classified within the Rhadinovirus genus, sharing evolutionary relationships with other gammaherpesviruses such as Kaposi Sarcoma-Associated Herpesvirus (KSHV, also known as Human gammaherpesvirus 8) and Murid gammaherpesvirus 4 (MuHV-4) [12]. The genomic analysis revealed a 103,374 base pair genome encoding 75 predicted ORFs, including 12 distinct genes [12]. GPHLV’s predicted open reading frames (ORFs) also shared homology with those of KSHV; specifically, 84% (63 out of 75 ORFs) showed significant sequence similarity [12]. 

Building on this foundation, we present the complete genome of GPXV along with a comparative analysis against GPHLV, as a reference for future studies investigating this virus and its role in gammaherpesvirus infections and pathogenesis. For phylogenetic analysis, ORFs 8, 9, 50, and 73 were selected due to their known roles in viral entry (gB), DNA replication (DNA polymerase), lytic reactivation (ORF50), and latency (ORF73), and because they are conserved across gammaherpesviruses, enabling reliable comparison. We hypothesize that GPXV shares genomic and functional similarities with various gammaherpesviruses such as Bat gammaherpesvirus 2, Marmot herpesvirus 1, and Saimiriine gammaherpesvirus 2. Furthermore, we hypothesize that ORF8, 9, 50, and 73 exhibit evolutionary conservation with homologous proteins, reflecting their roles in viral replication, immune evasion, and host adaptation.

## 2. Materials and Methods

### 2.1. Cell Lines and Viruses

Embryonic guinea pig 104C1 cells were sourced from the American Type Culture Collection (ATCC, Catalog Number: CRL-1405). These cell lines were cultured in RPMI-1640 (Gibco, Catalog Number: 11875) and supplemented with 10% fetal bovine serum (Hyclone, Catalog Number: SH30071.03HI) and 1 × Penicillin–Streptomycin (Hyclone, Catalog Number: SV30010), which provides a final antibiotic concentration of 100 U/mL of penicillin and 100 µg/mL of streptomycin. The cultures were maintained at 37 °C with 5% CO_2_, following the recommendations of the ATCC. Guinea Pig X Virus was acquired from the ATCC (ATCC, Catalog Number: VR-914) and cultivated in 104C1 cells.

### 2.2. Viral Purification and DNA Extraction

GPXV was propagated in 104C1 cells to high titers, defined as concentrations sufficient to produce robust viral replication, and titrated using a plaque assay to confirm viral titer. After an 8-day infection period, leading to a complete cytopathic effect, the cultures were frozen at −80 °C and later thawed at room temperature. The culture medium was collected in 50 mL tubes, and cell debris was pelleted by centrifugation at 3000× *g* for 10 min at 4 °C. This pellet was freeze-thawed twice more and resuspended in the supernatant. Viral purification involved ultracentrifugation [17]. Supernatants were clarified by centrifugation at 9000× *g* for 15 min at 4 °C and then filtered through a 0.45 µm syringe filter. The filtered supernatant was layered onto a 15% sucrose cushion and centrifuged at 80,000× *g* for 60 min at 4 °C. The viral pellet was resuspended in a minimal volume of phosphate-buffered saline (1 × PBS, pH 7.4) overnight at 4 °C. Subsequently, DNase I (NEB, Catalog Number: M0303S) treatment was carried out for 15 min at 37 °C to eliminate extraneous host DNA, which was then inactivated with 0.5 M EDTA to a final concentration of 5 mM and heat-inactivated at 75 °C for 10 min. The virus was lysed by adding 10% SDS to a final concentration of 1% PBS, along with RNase A (NEB, Catalog Number: T3018L) and Proteinase K (NEB, Catalog Number: P8107S) at 56 °C for 5 min. Phenol-chloroform extraction and ethanol precipitation were then used to isolate the viral DNA. This initial extraction enriched encapsidated viral DNA while minimizing host genomic contamination.

### 2.3. Generation of a Consensus GPXV Genome

Genomic DNA was then extracted from 104C1 cells infected with GPXV using the Qiagen DNeasy Blood and Tissue Kit (Qiagen, Catalog Number: 69504). This secondary extraction allowed for the targeted amplification of specific genomic regions via polymerase chain reaction (PCR). To address gaps identified in the initial genome assembly of the GPXV, we used a targeted approach using standard PCR and Sanger sequencing [18]. Gap regions were selected based on repetitive or ambiguous sequences, including the repetitive sequence in ORF73; the intergenic regions between ORF69 and ORF72, ORF56 and ORF57, and ORF10 and ORF17.5; and genomic regions between GPXV-specific genes G1 and G4, located 3′ to the ORF4 orthologue. The initial genome assembly, derived from high-throughput sequencing data, often contains regions of incomplete or ambiguous sequences due to limitations in coverage and sequencing errors. To overcome these gaps, we designed specific PCR primers flanking the gaps based on the consensus sequence generated during the assembly process [19]. These primers were designed to amplify the regions surrounding the gaps, making sure that the DNA segments of interest could be accurately captured and analyzed.

Standard PCR was performed using the genomic DNA extracted from the GPXV-infected 104C1 cells as a template. The reactions were optimized for annealing temperatures and reaction conditions to ensure high specificity and yield. Following successful amplification, the PCR products were purified to remove any residual primers and contaminants. The purified PCR products were then subjected to Sanger sequencing, a method known for its accuracy and reliability in sequencing individual DNA fragments [20,21]. The sequencing data obtained from Sanger sequencing was used to manually inspect and fill the gaps in the genome assembly. This process involved aligning the Sanger sequence data with the existing assembly to integrate new information and resolve ambiguities in the consensus sequence.

### 2.4. Next-Generation Sequencing, Viral Classification, De Novo Assembly, and ORF Identification

Whole-genome sequencing of GPXV was performed using the Illumina MiSeq platform with a 600-cycle MiSeq Reagent Kit v3. DNA concentrations for library preparation were measured using the Qubit dsDNA HS Assay Kit. Libraries were prepared from 1 ng of DNA per sample using the Nextera XT DNA Library Prep Kit and Nextera XT Index Kit v2 for sample barcoding. The quality, size, and concentration of the final libraries were verified using the Fragment Analyzer Instrument. Libraries were pooled in equimolar ratios and re-validated on the Fragment Analyzer before sequencing. Raw paired-end FASTQ reads were analyzed using the Bacterial and Viral Bioinformatics Resource Center (BV-BRC) Taxonomic Classification Tool (https://www.bv-brc.org/app/TaxonomicClassification; accessed on 29 November 2023).

For de novo genome assembly, FASTQ files were processed with Trimmomatic for quality control and filtering, using parameters of 36, 3, and 3 bases for minimal length, leading, and trailing filtering [22]. Genomic assembly was carried out with Spades, selecting the longest contig (103,374 nucleotides) for subsequent analysis [23]. The longest contig from the de novo assembly was annotated using Prodigal gene prediction software (https://github.com/hyattpd/Prodigal; version 2.6.3) [21].

Protein sequences for ORFs 8, 9, 50, and 73 were extracted from the annotated genome (GenBank format) using Biopython (https://biopython.org/; version 1.84) and subjected to BLASTp (https://blast.ncbi.nlm.nih.gov/Blast.cgi; version 2.16.0) searches against the viral database in the Bacterial and Viral Bioinformatics Resource Center (https://www.bv-brc.org/app/Homology; accessed on 25 August 2024) and InterPro (https://www.ebi.ac.uk/interpro/; accessed on 1 September 2024) to identify homologous viral proteins. A minimum ORF size threshold of 300 bp was applied, corresponding to proteins of at least 100 amino acids. This threshold was chosen to balance sensitivity with the risk of false positives. Additional criteria for ORF identification included the presence of start and stop codons. To mitigate the risk of false positives from smaller ORFs, additional verification was conducted for sequences between 300 bp and 500 bp through manual curation and comparative analysis with closely related Herpesviridae genomes. The protein sequences, along with homologs from related viral genomes, were aligned using Clustal Omega (https://www.ebi.ac.uk/jdispatcher/msa/clustalo; accessed on 24 September 2024). Separate phylogenetic trees were first generated for each ORF alignment using MAFFT/Multiple Alignment using Fast Fourier Transform (https://mafft.cbrc.jp/alignment/server/index.html; accessed on 17 October 2024).

Because gene conservation and annotation quality vary among ORFs and across gammaherpesvirus genomes, we did not use a uniform reference set for the phylogenetic analyses. Instead, for each ORF, homologous sequences were selected based on BLASTp results for high-confidence alignments and relevant evolutionary comparisons. This gene-specific selection strategy avoids including distantly related or poorly annotated sequences that could reduce tree accuracy or inflate alignment noise. While genes such as gB and DNA polymerase are generally conserved across *Rhadinovirus* and *Lymphocryptovirus* genera, we observed variability in annotation completeness and sequence quality across available genomes. Therefore, we maintained a flexible, BLAST-driven selection approach for each ORF. A full list of viruses used for phylogenetic comparison with GPXV, including their ICTV formal names, common names, natural hosts, and accession numbers, is provided in Appendix A.

Furthermore, a comparative genomic analysis between GPXV and GPHLV assessed homologous relationships and divergence in protein-coding content. Annotated GenBank files for both viruses were used as input for the analysis. Coding DNA sequences (CDSs) and associated metadata (gene names, start and end positions) were extracted using Biopython (https://biopython.org/; version 1.84). Protein sequences from GPXV were aligned to those of GPHLV using BLASTp (https://blast.ncbi.nlm.nih.gov/Blast.cgi; accessed on 13 June 2025) with an e-value cutoff of 1 × 10^−5^ identify homologs. To accommodate naming inconsistencies between the two genomes, fuzzy string matching was applied to protein labels. Genomic coordinates, gene names, and percent sequence identity were retrieved for each matched pair. Matches with less than 95% identity were annotated as divergent. GPXV ORFs lacking identifiable homologs in GPHLV were designated as unique to GPXV, and vice versa.

A comparative annotation table was generated (Appendix A) summarizing the homologous relationships between predicted proteins in the two viral genomes. The table includes GPXV and GPHLV gene/protein names, their respective genome coordinates, percent identity, and classification as divergent or unique. This dataset was used to support the interpretation of lineage-specific genes and evolutionary divergence between the two viruses.

In addition to the phylogenetic and comparative genomic analysis, a table summarizing the functional and genomic characteristics of ORFs 8, 9, 50, and 73 within GPXV was created (Appendix A). Information was gathered from various sources, including UniProt (https://www.uniprot.org/; accessed on 24 October 2024), NCBI Conserved Domains (https://www.ncbi.nlm.nih.gov/Structure/cdd/cdd.shtml; accessed on 24 October 2024), BLASTp (https://blast.ncbi.nlm.nih.gov/Blast.cgi; accessed on 24 October 2024), InterPro (https://www.ebi.ac.uk/interpro/; accessed on 24 October 2024), and AlphaFold (https://alphafold.ebi.ac.uk/; accessed on 24 October 2024), to support the characterization of these ORFs.

## 3. Results

### 3.1. Genomic Features and Homology of GPXV

The GPXV genome (Figure 1) yields a complete genome sequence of 109,113 base pairs. The resulting genome consists of a single contiguous sequence with no additional contigs confidently attributable to GPXV. Other contigs produced during assembly were either host-derived or represented low-complexity sequences lacking viral features and were excluded from the final analysis. This genomic structure is homologous to other rhadinoviruses, placing GPXV within the Gammaherpesvirinae subfamily. The virus encodes 62 predicted ORFs. Of these, 60 (96.8%) appear divergent from their counterparts in GPHLV, while only 2 ORFs (3.2%) are highly conserved. None of the ORFS are unique to GPXV. In addition, the genome encodes several potential non-coding RNAs (ncRNAs) and repetitive features.

### 3.2. Phylogenetic Analysis of GPXV ORF8, 9, 50, and 73

The phylogenetic tree (Figure 2) constructed using glycoprotein B (gB) and homologous sequences shows that GPXV ORF8 forms its own branch, distinctly associated with a clade containing gammaherpesviruses and rhadinoviruses primarily from Macaca species, including Rhesus monkey rhadinovirus (H26-95), Macacine gammaherpesvirus 5, and Macaca mulatta rhadinovirus. These sequences form a separate grouping within the tree. Furthermore, sequences from Retroperitoneal fibromatosis-associated herpesvirus, Human gammaherpesvirus 8, and Marmot herpesvirus branch earlier in the phylogeny, indicating lower similarity to GPXV ORF8.

Similarly, the phylogenetic tree (Figure 3) constructed using the DNA polymerase catalytic subunit and homologous sequences indicates that GPXV ORF9 forms a distinct branch. It aligned distantly with sequences from Myotis ricketti herpesvirus 1, Saimiriine gammaherpesvirus 2, and Rhinolophus gammaherpesvirus 1. Bovine gammaherpesvirus 4 and Macaca fuscata rhadinovirus are positioned in an intermediate clade. Additionally, Marmot herpesvirus 1, Peromyscus leucopus gammaherpesvirus, and Saguinine gammaherpesvirus 1 form separate, branching clades. Human-associated Gammaherpesvirus 8 diverges from GPXV ORF9 within the same cluster but shows lower similarity.

Furthermore, the phylogenetic tree (Figure 4) constructed for GPXV ORF50 reveals that GPXV forms a cluster with Bovine gammaherpesvirus 4 and a distinctly related clade with transcription activation factors from Eptesicus fuscus gammaherpesvirus and Sea otter herpesvirus. The clustering pattern suggests that GPXV ORF50 is more closely related to gammaherpesviruses associated with terrestrial mammals, particularly those of bats (Eptesicus fuscus) and marine mammals (Sea otter herpesvirus). Marmot herpesvirus 1 and Saguinine gammaherpesvirus 1 group together in a separate clade. Murid herpesvirus and Wood mouse herpesvirus form another group at a greater phylogenetic distance from GPXV ORF50.

Lastly, a phylogenetic tree (Figure 5) was constructed using the ORF73 protein sequence from GPXV and homologous sequences from other members of the Gammaherpesvirinae subfamily. The phylogenetic tree shows that GPXV ORF73 forms a separate branch, away from other gammaherpesviruses included in the analysis. The human gammaherpesvirus 8 sequences, including ORF73 Human gammaherpesvirus 8, ORF73 F997A/F1001A Human gammaherpesvirus 8, latent nuclear antigen partial Human gammaherpesvirus 8, LANA Human gammaherpesvirus 8, and Herpesvirus saimiri ORF73 homolog Human gammaherpesvirus 8, cluster together, indicating their close relationship. The primate gammaherpesviruses, including ORF73 Macaca nemestrina rhadinovirus 2, latency-associated nuclear antigen Rhesus monkey rhadinovirus H26-95, latent nuclear antigen Macacine gammaherpesvirus 5, and ORF73 Macacine gammaherpesvirus 5, form a distinct clade. Felis catus gammaherpesvirus 1 ORF73 is positioned within the primate gammaherpesvirus clade. Murine roseolovirus hypothetical protein MRV_0112 is placed as an outgroup, showing the greatest evolutionary distance from the other viruses in the tree.

Overall, phylogenetic tree analysis was conducted using the predicted protein sequences of six well-conserved genes (uracil-DNA glycosylase, helicase-primase-primase helicase subunit, DNA packaging terminase subunit 1, major capsid protein, envelope glycoprotein B, and DNA polymerase catalytic subunit) across various members of the Gammaherpesvirinae subfamily, including GPXV. GPXV ORF73, encoding a latency-associated nuclear antigen (LANA), clusters closely with Human Gammaherpesvirus 8 ORF73 and latency antigens. More distantly related sequences include Felis catus gammaherpesvirus 1, Macaca nemestrina rhadinovirus 2, Rhesus monkey rhadinovirus, and Macacine gammaherpesvirus 5. GPXV ORF8, encoding the envelope glycoprotein B, clustered within a clade composed primarily of primate-associated gammaherpesviruses, including Macacine gammaherpesvirus 5, Rhesus monkey rhadinovirus H26-95, and Macaca nemestrina rhadinovirus. GPXV ORF9, with its DNA polymerase catalytic subunit, grouped with sequences from Bovine gammaherpesvirus 4 and Peromyscus leucopus gammaherpesvirus, forming a distinct clade. GPXV ORF50, having the transcriptional activator, was positioned within a clade containing Bovine gammaherpesvirus 4 and Sea otter herpesvirus, separating it from both the primate- and non-primate-associated clades.

### 3.3. Genomic Comparison Between GPXV and GPHLV

A comparative annotation of the GPXV and GPHLV genomes identified both shared and unique protein-coding genes across the two viruses. Of the 62 predicted ORFs in GPXV, 60 (96.8%) appear divergent from their counterparts in GPHLV, while only 2 ORFs (3.2%) are highly conserved. Twenty-four non-ORF features were classified as unique, with no detectable homologs in the GPHLV genome. These unique features include a predicted vIL-10 homolog and several unnamed hypothetical proteins. Among the remaining GPXV ORFs, most had identifiable counterparts in the GPHLV genome but exhibited varying degrees of sequence divergence. Percent amino acid identity among matched gene pairs ranged from 19.35% to 91.67%. In particular, ORF39 and ORF43 were among the most highly conserved pairs between the two viruses, sharing over 90% identity, while several other gene pairs, including ORF30 and ORF52, showed lower identity, suggesting greater divergence. In addition to positional homologs, a number of genes were found to have shifted or have inconsistent annotations between the two genomes. For example, GPXV ORF20, ORF31, and ORF41 all corresponded to the same annotated region in GPHLV (ORF10), while GPHLV contained several unique genes (e.g., G10, G12, ORF29b, and RTA) that were not present in GPXV.

## 4. Discussion

The assembly of the GPXV genome reveals 62 predicted ORFs and 24 GPXV-specific non-ORF features. All 62 ORFs show variable homology to GPHLV (with percent identity ranging from 19.35% to 91.67%), placing GPXV within the gammaherpesvirus subfamily. However, the presence of 24 GPXV-specific features and their solitary placements within phylogenetic trees suggest unique functional genes and proteins that may contribute to the virus’s adaptation to its guinea pig host. GPXV contains a set of ORFs (Table 1) that may enhance its ability to evade host immune responses and establish latency, like KSHV, which utilizes specific ORFs for immune evasion and latency maintenance [14,24,25]. Certain ORFs in GPXV may encode proteins with specialized functions that modulate host cell signaling pathways, thereby improving the virus’s capacity to persist within the host [7]. While all herpesviruses share fundamental replication strategies, GPXV may employ specific adaptations that optimize its replication cycle in guinea pigs, involving interactions with host cellular machinery or alternative pathways for genome replication that distinguish it from other herpesviruses [7,26,27].

The phylogenetic analysis of ORF8 from GPXV shows its close relationship with glycoprotein B sequences from primate-associated gammaherpesviruses, including Macacine gammaherpesvirus 5 and Rhesus monkey rhadinovirus H26-95. As glycoprotein B is a well-conserved entry protein across herpesviruses, this clustering is consistent with its role in viral infection and transmission [28,29]. The placement of GPXV’s ORF8 within this group suggests the conservation of structural or functional elements necessary for viral entry. Its phylogenetic distance from Human gammaherpesvirus 8 (Kaposi Sarcoma-Associated Herpesvirus) and Marmot herpesvirus 1 may indicate host-specific adaptations rather than a divergence in function.

Meanwhile, GPXV ORF9, which encodes the DNA polymerase catalytic subunit, has evolutionary proximity to several gammaherpesviruses associated with diverse mammalian hosts. GPXV clusters closely with Myotis ricketti herpesvirus 1 and Saimiriine gammaherpesvirus 2, suggesting a conserved functional role in viral replication across different host species. DNA polymerase is essential for viral genome replication, and its conservation across these species highlights its evolutionary importance [27]. The distant placement of GPXV from Human gammaherpesvirus 8 (Kaposi Sarcoma-Associated Herpesvirus) and Marmot herpesvirus 1 indicates divergence in the DNA polymerase sequence, possibly driven by adaptation to its guinea pig host. Interestingly, the clustering of GPXV with bat- and primate-associated gammaherpesviruses suggests that its DNA polymerase might retain core functionalities shared among these diverse hosts, while its distinct separation from rodent-associated viruses like Peromyscus leucopus gammaherpesvirus suggests species-specific evolutionary pressures.

Moreover, the phylogenetic analysis of GPXV ORF50 indicates that it has a close relationship with transcription activation factors of murine-associated gammaherpesviruses, particularly Murid gammaherpesvirus 4 and Wood mouse herpesvirus. ORF50 encodes a key regulatory protein, the replication and transcription activator (Rta), which initiates lytic replication by activating downstream viral promoters, emphasizing that GPXV may share similar mechanisms for transitioning from latency to lytic replication [30]. Interestingly, GPXV is positioned between rodent and non-rodent gammaherpesviruses, indicating a divergence from marine mammal and bat gammaherpesviruses, such as Myotis and Rhinolophus species. Its relatively distant placement from sea otter and bovine gammaherpesviruses shows potential host-specific adaptations in ORF50, aligning more closely with rodent and small mammal viruses. Additionally, GPXV’s divergence from Marmot and Saguinine gammaherpesviruses further emphasizes its distinct evolutionary path within the gammaherpesvirus subfamily.

The analysis of ORF73 reveals its solitary evolutionary placement, suggesting a novel lineage within the Gammaherpesvirinae subfamily. ORF73 encodes a LANA1-like protein, which likely contributes to viral latency by tethering viral episomes to host chromatin, regulating gene expression, and modulating host immune responses [31,32]. Its divergence from homologs in related gammaherpesviruses may reflect functional specialization [31,32]. The similarity to Human gammaherpesvirus 8 (HHV-8) LANA suggests that the GPXV protein plays a comparable role in episomal maintenance, transcriptional control, and immune evasion during persistent infection [33,34]. However, its significant distance from the tightly clustered Macaca-related gammaherpesviruses (Macacine gammaherpesvirus 5, Macaca nemestrina rhadinovirus 2, and Rhesus monkey rhadinovirus) suggests that GPXV ORF73 is divergent, suggesting sequence-level variation potentially linked to differences in host-specific factors. While the functional conservation of episome tethering is likely, GPXV ORF73 exhibits divergence in regions corresponding to known chromatin-binding and transcriptional regulatory domains in primate LANA proteins. These differences warrant further structural and functional investigation to determine whether they represent host-specific adaptations or lineage-specific divergence.

In addition, the comparative genomic analysis between GPXV and GPHLV revealed divergence in gene content and sequence identity, emphasizing the distinct evolutionary trajectories of these two gammaherpesviruses. While a majority of GPXV ORFs had identifiable counterparts in GPHLV, most exhibited moderate to high levels of divergence at the amino acid level, suggesting functional divergence or adaptation to different hosts or ecological niches. Twenty-four non-ORF features were identified as being unique to GPXV, with no detectable homologs in the GPHLV genome. These included a vIL-10 homolog and several uncharacterized proteins. The presence of a vIL-10-like gene in GPXV but not in GPHLV may suggest immune-modulatory adaptations specific to the guinea pig host, a pattern observed in other rhadinoviruses that encode cytokine mimics to evade host immunity [28]. Additionally, the abundance of uncharacterized genes and proteins unique to GPXV raises the possibility of lineage-specific gene gains or accelerated evolution, suggesting further research into its function.

The conservation of certain core herpesvirus genes, such as ORF39 and ORF43, which showed over 90% amino acid identity, supports the phylogenetic relatedness of GPXV and GPHLV within the gammaherpesvirus subfamily. However, the variable identity levels across other ORFs (ranging as low as 19.35%) highlight regions of the genome potentially under relaxed selective pressure or experiencing gene remodeling. Such divergence may reflect host-specific pressures or differences in tissue tropism, latency mechanisms, or immune evasion strategies [35]. In some cases, a single GPXV gene corresponded to multiple or re-annotated regions in GPHLV, such as ORF20, ORF31, and ORF41, all aligned to GPHLV ORF10. This suggests potential annotation inconsistencies, gene fragmentation, or structural rearrangements that further complicate direct comparisons between the two genomes. Similarly, GPHLV harbored unique ORFs (e.g., ORF29b, G10, and RTA) that were not present in GPXV, reflecting reciprocal patterns of gene presence/absence.

Together, these findings emphasize the genomic plasticity of rhadinoviruses and highlight GPXV as a genetically distinct herpesvirus within this group.

## 5. Conclusions

Guinea pigs have emerged as an important animal model for studying herpes simplex virus (HSV) infections [36]. This model closely mimics human disease, allowing the evaluation of vaccines and antiviral therapies [37,38,39,40]. Despite their usefulness, guinea pigs have limitations that make it difficult to fully understand the virus and its associated diseases. While various vaccine candidates have demonstrated protective effects in guinea pigs, the translation of these findings to human applications remains uncertain due to differences in immune system responses and viral pathogenesis [41,42].

In this context, we sequenced and analyzed the genome of GPXV, resulting in a 109,113 bp consensus sequence homologous to other rhadinoviruses, confirming its classification within the Gammaherpesvirinae subfamily. The genome encodes 62 predicted ORFs showing homology to GPHLV [23]. Phylogenetic analyses across multiple conserved genes (e.g., ORF8, ORF9, ORF50, and ORF73) reveal GPXV as a distinct lineage, diverging from both primate- and rodent-associated gammaherpesviruses. This divergence suggests host-specific adaptations, particularly in genes involved in latency, immune modulation, and replication [43]. Shared ancestry with viruses such as Peromyscus leucopus gammaherpesvirus and Macacine gammaherpesvirus further supports the functional conservation of key viral proteins.

While this study provides a comprehensive genomic characterization of GPXV and its phylogenetic placement within the Rhadinovirus genus, several limitations must be acknowledged. Our selection of viruses for individual ORF comparisons was based on BLASTp results. Since some ORFs are not well conserved across all seven genera within the subfamily, we selected the most relevant sequences for each ORF, rather than forcing all genera into each analysis. While this approach allows for the inclusion of the most relevant sequences, it may limit direct comparisons across the entire subfamily. Second, our study primarily focuses on genomic sequence analysis and does not include the functional validation of predicted ORFs or gene products. While we provide putative gene functions based on homology, further experimental studies will be necessary to confirm the biological roles of these genes in GPXV. Additionally, we acknowledge the absence of detailed functional annotations for certain genes, including the G gene family, due to limited data availability.

To address this gap, we conducted a comparative genomic analysis between GPXV and the previously described GPHLV. This direct comparison revealed divergence in sequence identity and gene content between the two viruses, including 24 non-ORF features in GPXV, such as a viral IL-10 homolog. These differences suggest lineage-specific adaptations potentially related to immune evasion or host specificity. The identification of both conserved and divergent genes across the two genomes shows GPXV’s genetic distinctiveness within rodent-associated gammaherpesviruses.

Collectively, these findings establish GPXV as a novel gammaherpesvirus with both conserved viral functions and unique genomic features adapted to its guinea pig host. This work provides a basis for future investigations into GPXV, including its functional genomics, host interactions, and potential utility as a model system for studying gammaherpesvirus evolution and pathogenesis.

## Figures and Tables

**Figure 1 viruses-17-01084-f001:**
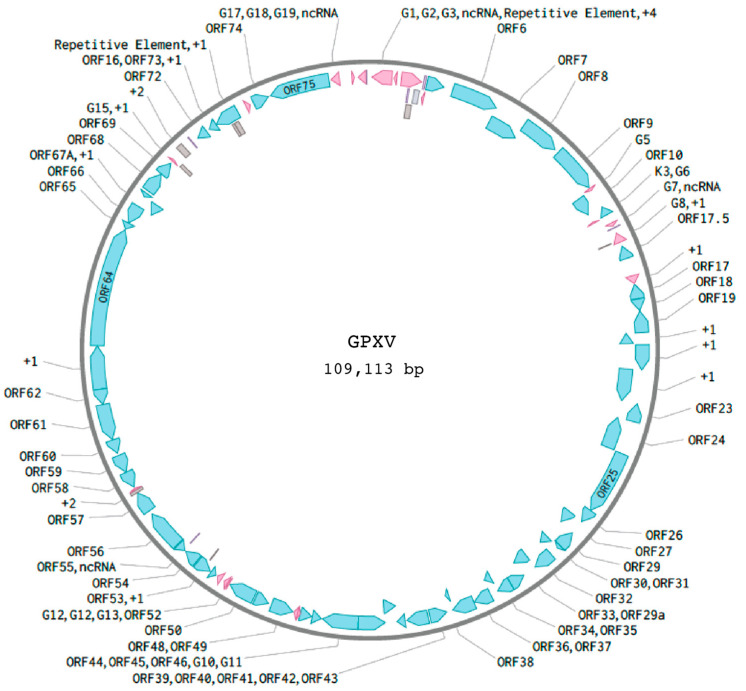
Schematic of GPXV genome. A 109.1 Kbp consensus genome was de novo assembled from Illumina NGS reads of DNA isolated from purified GPXV virions. The genomic structure of GPXV contains 24 GPXV-specific non-ORF features (pink). The GPXV genome encodes several potential ncRNAs (purple) and repetitive features (gray). Overlapping ORFs are labeled with “+1” or “+2” to indicate their reading frame relative to adjacent genes. These are independently predicted ORFs, not alternate transcripts or isoforms, and are included for completeness based on automated genome annotation pipelines.

**Figure 2 viruses-17-01084-f002:**
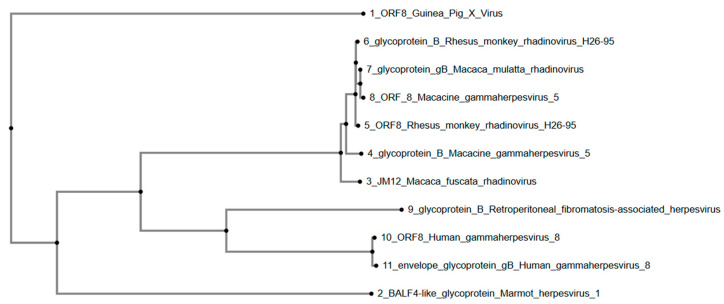
Phylogenetic tree of GPXV ORF8. GPXV does not cluster phylogenetically with other viruses in the gammaherpesvirus and rhadinovirus subfamilies of herpesviruses, and is an outgroup. Rendered from NCBI BLASTp results using MAFFT. Note: Human gammaherpesvirus 8 is also referred to as Kaposi Sarcoma-Associated Herpesvirus (KSHV).

**Figure 3 viruses-17-01084-f003:**
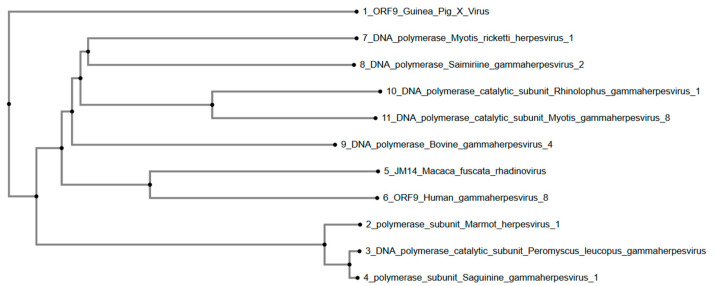
Phylogenetic tree of GPXV ORF9. Again, GPXV is positioned as an outgroup away from other rhadinoviruses and gammaherpesviruses. Rendered from NCBI BLASTp results using MAFFT.

**Figure 4 viruses-17-01084-f004:**
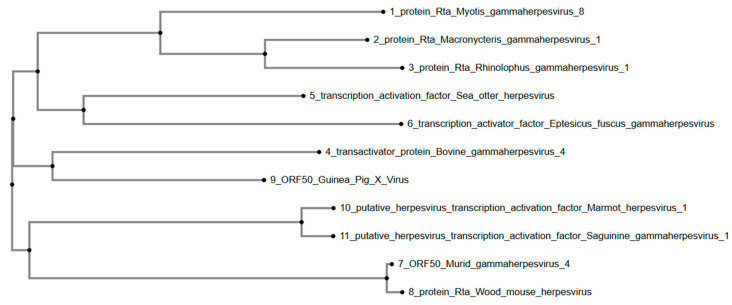
Phylogenetic tree of GPXV ORF50. GPXV clusters phylogenetically with Bovine gammaherpes 4. Rendered from NCBI BLASTp results using MAFFT.

**Figure 5 viruses-17-01084-f005:**
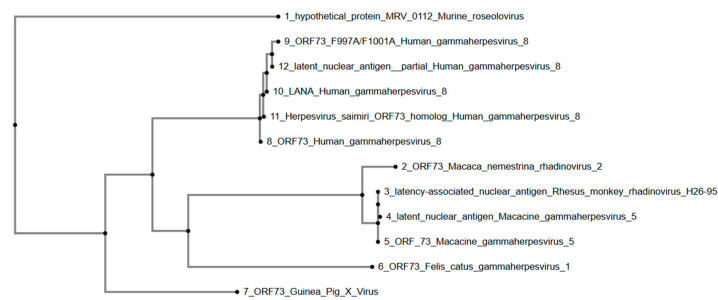
Phylogenetic tree of GPXV ORF73. Phylogenetic tree of ORF73 sequences showing Guinea Pig X Virus ORF73 as a solo branch. Human, primate, and feline gammaherpesviruses form separate clades, with Murine roseolovirus as an outgroup. Rendered from NCBI BLASTp results using MAFFT.

**Table 1 viruses-17-01084-t001:** Functional and genomic characteristics of ORF8, ORF9, ORF50, and ORF73 within GPXV. This table summarizes the functional domains, closest homologs, subcellular localization, E-values, and percent identity for four key open reading frames (ORFs) identified in GPXV. ORF8 encodes a glycoprotein B PH-like protein associated with viral receptor functions. ORF9 is a DNA polymerase involved in replication and nucleotide binding. ORF50 functions as a herpesvirus transcription activator. ORF73 encodes a LANA1-like protein with DNA-binding activity. The closest homologs include proteins from Marmot herpesvirus 1, Bovine herpesvirus 4, Saguinine gammaherpes virus 1, and Human gammaherpesvirus 8 with subcellular localization ranging from the host cell endosome to the nucleus.

ORF Name	Protein Name	Functional Domains/Motifs	Closest Homolog (Species)	Subcellular Location	E-Value	Percent Identity
**ORF8**	Herpesvirus Glycoprotein B PH-like	Facilitates virion assembly, egress, and entry into host cells by interacting with viral and cellular membranes.	Marmot herpesvirus 1, Saguinine gammaherpesvirus 1, KSHV (HHV-8)	Host cell endosome	0	56.93–59.80%
**ORF9**	DNA polymerase	DNA polymerase family B; This region of DNA polymerase B appears to consist of more than one structural domain, possibly including elongation, DNA-binding and dNTP binding activities.	Bovine herpesvirus 4 (BoHV-4), Marmot herpesvirus 1, Saguinine gammaherpesvirus 1	Host nucleus	0	58.57–58.76%
**ORF50**	Herpesvirus transcription activation factor	Transcriptional activator that initiates lytic replication by activating viral early gene expression This family includes EBV BRLF1 and similar ORF 50 proteins from other herpesviruses.	Marmot herpesvirus 1, Saguinine gammaherpesvirus 1, Bovine herpesvirus 4 (BoHV-4)	Host nucleus	1 × 10^−44^	38.78–39.02%
**ORF73**	Protein LANA1-like, DNA-binding domain	Anchors viral episomes to host chromosomes and regulates latency gene expression	Human gammaherpesvirus 8	Host nucleus	5 × 10^−6^	37.80–38.27%

## Data Availability

Original FASTQ files for this study are available upon request. The consensus genome for GPXV has been deposited in the NCBI Nucleotide collection under the GenBank Accession Number PV335591.1.

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
