# Peer review of "Guinea Pig X Virus Is a Gammaherpesvirus"

_viruses, 2025, doi:10.3390/v17081084_

Round 1

Reviewer 1 Report (Previous Reviewer 3)

Comments and Suggestions for Authors

The revised manuscript (Truong et al. ) focuses on dissecting the genome contents and reveal phylogenetic divergences of gammaherpesviruses infecting guinea pigs. The significance of the study is largely improved by the new addition of a direct comparison of GPXV to GPHLV.  The revisions described the findings from the new comparison, and provided description of the newly included comparative study and revised the text as needed. Importantly, the authors reveal unique and shared genes present in the two related viruses, while also compare the viral genes to human GHV such as KSHV. The authors answered the key concerns of the reviewers, and adequately revised the manuscript. 

Author Response

Reviewer 2 Report (New Reviewer)

Comments and Suggestions for Authors

This paper reports the results of guinea pig X virus genome nucleotide sequencing and analysis. As for the paper itself, although it is redundant as a whole, the necessary information is not described, and it is not sufficient from a scientific point of view. Introduction and Conclusion are very long, and these need to be rewritten to a more concise content. Two types of DNA extraction are described for the method, but it is unclear how each was used. Regarding the PCR method used for DNA amplification, it is unclear which part was amplified using which primers. In the analysis, it is said that the protein sequence was extracted from the raw data, but without doing this, it should be possible to respond by extracting it from the determined genome sequence. In addition, phylogenetic analysis is carried out, but there is no description of what program was used, and it is impossible to confirm the validity as a phylogenetic tree. Several ORFs have been used for detailed analysis, but there is no scientific explanation for why these were chosen. Overall, there are only abstract descriptions, and no specific data is shown. Therefore, this paper is not considered scientifically prepared.

Author Response

Reviewer 3 Report (New Reviewer)

Comments and Suggestions for Authors

The authors have sequenced a guinea pig herpes virus (GPXV) and posted the 'largest config' onto the NCBI genome website (PV335591.1). With this sequence the authors denote the ORFs predicted by Prodigal gene prediction, and then compare a few of these genes to other herpes viruses using BLASTp. While the premise is simple, it is an important publication for a virus that has seemingly been used in guinea pig virus research but had never been fully sequenced or annotated. It would be good to know if the authors believe the entire genome is 109113 bp or if there are other contigs that were sequenced that were not included in the publication.

Author Response

This manuscript is a resubmission of an earlier submission. The following is a list of the peer review reports and author responses from that submission.

Round 1

Reviewer 1 Report

Comments and Suggestions for Authors

In this resubmission, the authors have added additional text clarification, literature, and an additional phylogenetic analysis of ORF73.  These have greatly improved the text. While this paper relies heavily on phylogenetic analysis of conserved, sequence-similar ORFs that can generate differing family trees, such analyses have been performed using standard tools. This is ultimately the best available option for analyzing proteins from limited numbers of representative strains and for viruses that have had millions of years to adapt to their specific hosts, and make ancestral inheritance difficult to distinguish from convergent evolution. Nevertheless, the work provides a full genome which should soon be available on GenBank after publication for a virus in a major animal model. My primary comments have been addressed, and I have only two minor comments for text edits:
Line 193: "De Novo" should be lowercase
Section 6 for patents can be removed

Reviewer 2 Report

Comments and Suggestions for Authors

Major concerns:

  1. The authors state: I also do not see the need to specify which GPXV gene encodes vIL-10, as it is not the focus of the study.

->This is a really concerning push back. If they are publishing the structure and genetic content of a new gammaherpesvirus genome, and they refer to a vIL-10 in the other guinea pig herpesvirus genome they published in lines 101 -103, it is obvious and expected that the authors would identify the vIL-10 gene name in the GPXV genome. They state that ‘We hypothesize that GPXV shares genomic and functional similarities with Bat gammaherpesvirus 2, Marmot herpesvirus 1, and Saimiriine gammaherpesvirus 2, particularly in the encoding of a viral IL10 homologue.’ Why is there a long paragraph dedicated to vIL-10 in the discussion in lines 365-383 if there is not data in the results that describe a vIL-10 in GPXV.  

When I analyzed their .gb file I see a cds annotation describing a vIL10 at coordinates 22,498 -> 23,016. Why are they reticent to describe this in their results?

  1. The genome should be available for reviewer review and not be held back until acceptance. I can not effectively review this manuscript without that data. It is bewildering that they can’t provide a detailed map and/or table of genome coordinates and ORFs/ cis-elements (repeats or non-coding regions) in the paper. This is a standard feature of new herpesvirus genome reports. What are the unique genes, and is vIL-10 one of them?
  2. EVERY phylogenetic tree for each ORF should contain the SAME genomes and should span human, primate and rodent gammaherpesviruses that represent commonly used lab strains and model pathogens used in the gammaherpesvirus field.
  3. With regard to redundancy, take note of these three sentences that have the same phrases repeated three times.

ORF73 encodes a LANA1-like protein that plays critical roles in viral latency, genome maintenance, and immune evasion, and its divergence in GPXV may indicate specialized adaptations [31,32]. GPXV is closely related to the LANA proteins and latent nuclear antigens of Human gammaherpesvirus 8, suggesting shared functional features in episomal genome maintenance, host chromatin tethering, transcriptional regulation, and immune evasion. Like HHV-8 LANA, the GPXV LANA-like protein may facilitate viral latency by tethering viral episomes to host chromosomes, regulating viral and host gene expression, and modulating immune signaling pathways to evade host immune responses

5.This statement is restating the obvious outcome of co-evolution of viruses with their hosts:

“However, its significant distance from the tightly clustered Macaca-related gammaherpesviruses (Macacine gammaherpesvirus 5, Macaca nemestrina rhadinovirus 2, and Rhesus monkey rhadinovirus) suggests potential host-driven evolutionary pressures, potentially shaped by species-specific differences in immune responses, viral latency strategies, or cellular interactions. The restricted presence of LANA proteins in humans and nonhuman primates highlights the evolutionary constraints within primate hosts while emphasizing the adaptability of gammaherpesviruses in optimizing long-term persistence within their specific host lineages”.

There is nothing learned from this restatement of well-known dogma in the field. ALL rhadino-herpesviruses have a ORF73/LANA protein to maintain their episome. The interpretation that ORF73 is similar but different so therefore it is conserved but divergent is obvious and lacks insight. A guinea pig rhadinovirus will OF COURSE differ from human and primates. Every herpesvirus evolves with its host- therefore a homolog will maintain function (like tethering episomes) while showing some host adaptations and natural variations as millions of years pass and species evolve.

6. They state: These include uracil-DNA glycosylase, helicase-primase helicase subunit, DNA packaging terminase subunit 1, major capsid protein, envelope glycoprotein B, and DNA polymerase catalytic subunit. Our methodology follows this principle, and we have selected different sets of viruses for individual ORFs based on their BLASTp results. Since some ORFs are not well-conserved across all seven genera within the Gammaherpesvirinae subfamily, we include the most relevant sequences rather than forcing all genera into each analysis. Given this approach, we will maintain our current strategy rather than using a uniform set of viruses across all ORF analyses.

Please note that gB and DNA pol are absolutely conserved across the rhadino and lymphocryptovirus genera within the gammaherpesviruses, that are most relevant. A uniform set spanning human, primate and rodents should be used across all ORF analyses for gB, DNA pol along with the other ORFs the author has chosen to focus on.

7. With respect to the guinea pig herpes-like virus, our study is specifically focused on GPXV. While we acknowledge the previously described guinea pig herpes-like virus, we have referenced the original study rather than conducting a direct comparison, as a full comparative analysis falls beyond the scope of this manuscript.

This is unacceptable, there are two herpesviruses derived from guinea pigs that are gammaherpesviruses. Are these lympho or rhadino-? How similar are they- this is STANDARD for new genomes, you simply compare to reference genomes and genomes from most closely related hosts.  Why does the genbank accession # Caviid gammaherpesvirus 2 isolate GPXV from a JV paper in 1980? refer to https://pubmed.ncbi.nlm.nih.gov/6255209/

A direct comparison is warranted here- how else can the author refer to the 1980 publication if he does not know if they are similar viruses? The origination of all GPXV should be detailed here.

8. They state that they provide no functional predictions available for the G genes, and we have not included speculative interpretations.

A blastp for conserved motifs is NOT speculative- this is using the databases to identify novel homologs that might identify unique host adaptations.

9. They clarify that The ‘+1’ and ‘+2’ notations in Figure 1 indicate overlapping reading frames rather than additional unlabeled genes.

Overlapping/alternate reading frames should NOT be included. These are completely different proteins and make no sense to include.

10. “In response to the concern about vIL-10, we acknowledge the importance of understanding its evolutionary significance. However, identifying the specific GPXV gene encoding vIL-10 is not the primary focus of this study. Our discussion on vIL-10 is meant to highlight broader evolutionary patterns observed in gammaherpesviruses rather than conduct a gene-level characterization within GPXV.”

 The authors also include this text: “This is an evolutionary linkage between viruses carrying the vIL-10 gene. Gammaherpesviruses, such as Epstein-Barr virus and Macaque CMV (RhLCV), have vIL-10 genes with remarkably high sequence identity to their host cellular IL-10 (cIL-10), with Epstein-Barr virus vIL-10 having 92% amino acid identity to cIL-10 and Macaque CMV (RhLCV) vIL-10 having 97% amino acid identity to cIL-10 [28]. This high degree of conservation suggests that the gene capture event occurred relatively recently in the evolutionary history of gammaherpesviruses, allowing less time for divergence compared to other herpesvirus subfamilies”.

This makes no sense. Is there a GPXV gene encoding vIL-10 or not? If there isn’t one, then it should not be discussed at all, anywhere in the paper. I see an annotation for one in the .gb supplemental file at genome coordinates 22,498 -> 23,016. What is the identity to all the vIL-10 and host IL-10 highlighted in this paragraph?

11.         It is remarkable that the Genbank citation states: JOURNAL   Viruses (2025) In press. That is presumptuous and untrue. Why is there a comment stating that ‘GenBank staff is unable to verify sequence and/or annotation provided by the submitter.’ 

A complete annotated genome needs to be available to review this data prior to acceptance. It is unethical to state that it is already in press.

12. The accession numbers for the viral genomes used in phylogenetic analysis and listed in supplemental tables should be provided.

Reviewer 3 Report

Comments and Suggestions for Authors

This Truong et al. manuscript provides some novel findings related to the genome and conservation of guinea pig x virus (GPXV), which would be valuable for the virology field. However, in this current form the provided comparisons, conclusions and the impact are somewhat limited. The manuscript lacks several key information, which would largely increase the scientific robustness, clarity and impact of the findings.  

The study would be of a much higher value to the field, if it would provide some additional key information, such as a comprehensive table with a list of coordinates of all the described genes with annotations, sequences included.

Furthermore, as the GPXV virus is closely related to other recently described GP viruses, such as the GPHLV virus (described similarly by the senior author’s group), yet no direct comparison is provided to these in the figures. To increase the significance of the study, and to provide more useful information to the field, the authors should include such a comparison in some form.  This could be a comparison of the genome, gene/protein. Best would be a table format, describing proteins encoded by the 2 viruses, their conservation between the closely related viruses. This would be highly valuable, as it would show differences/similarities and introduce a highly valuable comparison of these viruses.  

Additional notes:

-The value of the concatenated comparison on Fig 6 is not clear. Additionally, the content of Figure 6 is not easily read or interpreted. The individual comparisons already describe the same content. The reviewer recommends removing this figure.

-Large sections focus on vIL-10 such as end of Introduction (Line 121), as well as a longer paragraph in Discussion (Line 365-383). However, no findings were described about vIL-10 in this study. Authors should reveal vIL-10 in GPXV virus genome as well, or, alternatively, majorly shorten this section in discussion to a sentence or two. The extensive vIL-10 long description does not relate to this study and dilutes other findings described in the discussion.

-KSHV name should be corrected to “Kaposi’s Sarcoma Associated..” (such as in Line 110). This should be checked throughout the manuscript, for consistency with the KSHV field.
